# Determinants of unintended pregnancy and induced abortion among adolescent women in Ethiopia: Evidence from multilevel mixed-effects decomposition analysis of 2000–2016 Ethiopian demographic and health survey data

Tiruwork Amare[1], Fasil Tessema[2], Tamrat Shaweno[3]*

1 MSI Ethiopia, Reproductive Choices, Addis Ababa, Ethiopia, 2 Jimma University Institute of Health Department of Epidemiology, Jimma, Ethiopia, 3 Africa Centres for Diseases Control and Prevention, Addis Ababa, Ethiopia

* babiynos@gmail.com

**Data Availability Statement:** For this analysis, the data was accessed from the DHS Program website

## Abstract

### Background

Adolescents are highly at risk of unintended pregnancy due to physiological, sexual, social and psychological growth. The pregnancy may end with early childbirth, induced abortion and its complications. Although, the trends of unintended pregnancy and induced abortion have declined over time in Ethiopia, evidence is limited on key determinants for decline in order to propose vital areas of interventions. The current study aimed to identify the determinants of unintended pregnancy and induced abortion among adolescents over the decades.

### Methods

Trends in the prevalence of unintended pregnancy and induced abortion among adolescent women aged 15–19 years were investigated based using a series of the Ethiopia Demographic and Health Surveys (EDHS) data for the years 2000, 2005, 2011, and 2016. Subsample of adolescent women data was extracted from each survey. The combined datasets for unintended pregnancy and induced abortion over the study period (2000–2016) was analyzed. The percentage changes of trends of unintended pregnancy and induced abortion with its corresponding 95% CI for each variable were calculated. Multilevel mixed-effects decomposition analysis was applied to identify factors significantly associated with trends of unintended pregnancy and induced abortion among adolescents.

### Results

The trends of unintended pregnancy and induced abortion significantly declined during the study period. Unintended pregnancy among Ethiopian adolescents aged 15–19 years significantly decreased from 307 (41.4%) (95% CI: 35.7, 47.2%, p<0.001) in 2000 to 120 (25.1%)

(https://dhsprogram.com/data/available-datasets.cfm) for free. We do not have special access privileges to this data. All authors can access the data from this website. To get the data, researchers should register and log in. After logging into the DHS program website, independent researchers can access the data and submit a brief justification for the release of the data that includes an overview of their intended study.

**Funding:** Didn't receive any fund.

**Abbreviations:** CAC, Comprehensive Abortion Care; CSA, Central Statistics Agency; DHS, Demographic and Health Survey; EDHS, Ethiopia Demographic and Health Survey; HAPCO, HIV/AIDS Prevention and Control Office; MDG, Millennium Development Goals; PHCCO, Population and Housing Census commission Office; SRH, Sexual and Reproductive Health.

(95% CI: 18.9, 31.4%) in 2016. On the other hand, induced abortion significantly decreased from 62 (8.3%) (95% CI: 5.2, 11.4%) in 2000 to 20 (4.1%) (95% CI: 1.3, 6.9%, p = 0.004) in 2016. Age older than 18 years (Coeff = -0.41, 95%CI, -0.64, -0.18, p<0.001), living in Somali regional state (Coeff = -2.21, 95%CI, -3.27, -1.15, p<0.001) and exposure to media (Coeff = -0.60, 95%CI, -0.87, -0.33, p<0.001) showed a significance association with decline in unintended pregnancy whereas; living in Benshangul-Gumuz regional state (Coeff = -0.17, 95% CI, -0.32, -0.19, p = 0.03) and ANC service utilization history (Coeff = -0.81, 95%CI, -1.45, -0.17, p = 0.01) showed significance association with decline in induced abortion.

## Conclusion

The trends of unintended pregnancy and induced abortion significantly declined over the past decades in Ethiopia. Adolescent girls aged 17 years and above, exposure to media and living in Somali showed significant association with decline in unintended pregnancy whereas; living in Benshangul-Gumuz and ANC service utilization history showed significant decline with induced abortion. Exposure to media and utilization of Antenatal care (ANC) services may improve adolescent girls' reproductive health uptake.

## Introduction

Unintended pregnancy is a pregnancy that is either unwanted, such as the pregnancy occurred when no children or no more children were desired [1]. Although, unintended pregnancy sharply fell from 1990–2014 globally, it didn't show sharp declining trend in developing regions [2]. Globally, unintended pregnancy was 44% in 2014 and 65% of this burden happened in developing regions [3]. The magnitude of unintended pregnancy was highest in Africa and it was 24.0, 30.7, 35.9, 64.3% and 31.4 percent in Kenya, Egypt, Nigeria, South Africa and Ethiopia, respectively [3,4].

Induced abortion is the termination of a fetus that brought about intentionally using different options safe or unsafe based on quality of care [5,6]. Global estimates from 2010–2014 demonstrates that 45% of all induced abortions are unsafe and developing countries bear the burden of 97% of all unsafe abortions [5,6]. Abortion rates have declined significantly since 1990 in the developed world but not in the developing world [5]. In many areas of the world where rates of unintended pregnancy are high; unsafe abortions have also been shown to be high [7]. In Malawi, over 67% pregnancies are unintended and that of 27% ends with abortion [8]. Similarly, higher rate of abortion was reported in Nigeria [9] and in Congo [10].

According to World Health Organization (WHO)'s new study conducted in 36 countries, one in four pregnancies end up with unintended pregnancies and this was regarded as high [11]. Results from EDHS 2016 report indicate that unintended pregnancy and induced abortion are still high [12] and needs an ongoing priority intervention. Different study findings indicate that knowledge about contraceptive use, access to contraceptives, failure in contraceptive, sexual violence and level of decision on contraception choice can affect trends of unintended pregnancies and induced abortion among women. In addition, age and lower socioeconomic status were also documented predictors of unintended pregnancies and induced abortions [2,3,6,8].

Information on trends of unintended pregnancy and induced abortion has a great role in developing strategies and prevention modalities to minimize complications emanating from

unintended pregnancy and unsafe abortion. Although, there were studies conducted so far related to magnitude of unintended pregnancy and induced abortion among reproductive age women in Ethiopia [1–3], information is limited related to the determinants of trends of unintended pregnancy and induced abortion among adolescent women in Ethiopia. In addition, the magnitudes and determinants in different studies varied depending on the characteristics, study design and method of analysis. Moreover, information is limited on types of variables significantly playing role in decline or increment of unintended pregnancy and induced abortion among adolescents in Ethiopia. In areas with limited resource, identifying the key determinants of unintended pregnancy and induced abortion would be helpful for decision makers and policy analysts. Thus, this study aimed to determine the key predictors of unintended pregnancy, induced abortion and associated factors among adolescents in Ethiopia from 2000 to 2016.

## Methods

### Data source

The study utilized data from four consecutive Ethiopian Demographic and Health Surveys of 2000, 2005, 2011 and 2016 women's data (IR data is the EDHS women's data file code) set. According to the projected Ethiopian population report; by the year 2018 the estimated population of Ethiopia were 102.7 million of this 24.2% were adolescent and 11.7% of them were 15 to 19 year old. Adolescent girls were 5.8% from the total population [13]. The fertility rate was 56.6 births per 1000 women aged 15–19 years old. Until now, four demographic and health surveys were conducted in Ethiopia where reports are available in CSA online [12]. The first survey was conducted in 2000/1 followed by the 2005/6 EDHS which is part of the worldwide measure DHS project funded by USAID and was conducted under the auspices of the ministry of health and implemented by the then Population and Housing Census Commission Office (PHCCO). The EDHS surveys of 2011 and 2016 are also the same surveys like other years [12].

The EDHS used a two-stage stratified cluster sampling technique to select the study participants. In stage one, after each administrative region was stratified into urban and rural strata, Enumeration Areas (EAs) were selected using a probability proportional to EA size. In stage two, a household listing operation was carried out in all of the selected EAs and a fixed number of households from each EA were selected. All women aged 15–49 years who were permanent residents or who spend the night in the selected households the night before the survey were included [12]. In this analysis, a total weighted sample of 2,544 adolescent women (15–19 years old) who gave birth in the 5 years preceding the surveys or pregnant during the surveys or women who had unfavorable pregnancy outcomes were included. Trends in the prevalence of unintended pregnancy and abortion were investigated based on a series of the EDHS data for the years 2000 (n = 741), 2005 (n = 708), 2011 (n = 616), and 2016 (n = 479). Data from the surveys were appended together after extracting relevant variables for trend and multilevel mixed-effects decomposition analysis.

### Study variables

The dependent variables considered were trends of unintended pregnancy and induced abortion. Unintended pregnancy was defined as a pregnancy which is a sum of mistimed pregnancy (pregnancy wanted at a later time) and unwanted pregnancy (pregnancy which is not wanted at all) [3,7]. Abortion was taken from the EDHS question 'have you ever had a pregnancy termination?' with the response 'yes' if the woman ever had an abortion and otherwise 'no' as a binary outcome [12,14]. We used a simple Univariate filter, cross-tabulations, clinical knowledge and previous other related literature of interest; whether to decide which variables

**Table 1. List of selected variables for the decomposition analysis of trends of unintended pregnancy, induced abortion and associated factors among adolescents in Ethiopia using 2000–2016 EDHS data.**

| Unintended pregnancy | | Induced abortion | |
|---|---|---|---|
| Socio demographic and economic variables | Reproductive history related variables | Socio-demographic and economic variables | Reproductive history related variables |
| Age, residence, educational status, religion, employment status, household wealth, region of residence, family size, household, decision exposure to media | ANC visit history, place of delivery, abortion, mode of delivery, birth order, pregnancy plan, knowledge about family planning, age at first sex, ever use of family planning | Age, residence, educational status, religion, employment status, household wealth, region of residence, family size, household, decision exposure to media | ANC visit history, place of delivery, mode of delivery, birth order, pregnancy plan, knowledge about family planning, age at first sex, ever use of family planning |

to keep in the analysis or not. The selected explanatory variables used in this analysis are indicated below [Table 1].

## Data management and analysis

First, description of the study participants by socioeconomic, demographic and reproductive health services was made with frequencies and percent. This was followed by the estimation of the prevalence of unintended pregnancy and induced abortion by socioeconomic and reproductive health variables across survey years (2000, 2005, 2011 and 2016). Then, percentage point change with corresponding 95% CI of the outcome variables was calculated by each of the study factors including socio-economic, demographic and reproductive factors: to examine the changes over the survey years, 2000 to 2016. We used the combined dataset in order to detect any association between the study factors and the outcomes, as well as to examine trends in unintended pregnancy and abortion over the study years (2000–2016). Outcome variables with missing information (information missing on unintended pregnancy and abortion) were excluded from the study whereas; due to the cross sectional nature of the DHS survey, explanatory variables with greater than 5% missing value were excluded from further analysis. We estimated P for trends in each category of the study variables to assess whether the prevalence decreased or increased over the study period using chi-squared test for trend.

A multi-level mixed-effects decomposition logistic regression analysis of change in unintended pregnancy and induced abortion was applied to know the determinants for the changes in the trends of unintended pregnancy and induced abortion. The purpose of applying multi-level mixed-effects model was to take into account for the hierarchical nature of the DHS data whereas; application of decomposition analysis was to identify the sources of change in unintended pregnancy and induced abortion over the study periods. Bivariable multilevel mixed effects decomposition logistic regression analysis was carried to determine the coefficients at 95% confidence interval and those variables with p-value <0.25 were considered for multivariable decomposition analysis. In the multivariable multilevel mixed effects decomposition logistic regression analysis, those variables with p-value <0.05 were considered as significantly associated with outcome variable. The Intra-Class Correlation (ICC) was estimated to assess the cluster effect and the model fitness was compared using information criteria (IC). All statistical analyses were conducted using Stata version 18.0 with 'svy' command to adjust for sampling weights, clustering effects and stratification.

## Ethics statement and consent to participate

Ethical approval is not applicable for analysis and reporting from part of the EDHS data which is available upon reasonable request from the system. However, ethical clearance for the 2016 EDHS was provided by the Ethiopian Health and Nutrition Research Institute (EHNRI) Review Board, the National Research Ethics Review Committee (NRERC) at the Ministry of

Science and Technology, the Institutional Review Board of ICF International, and the communicable disease control (CDC). We confirm that the study was conducted in accordance with the Helsinki Declaration. Written consent was obtained from each participant. In addition, parents/legal guardians were consented for minors aged 15 and 16 years. The parental consent for minors was also approved by the above mentioned national and international ethical bodies [12]. However, the dataset of the 2016 EDHS is not available as a public domain survey dataset. The author requested access to the data from demographic, health survey program team and access was granted to use the data for this research. Authorization letter was taken from http://www.measuredhs.com.

## Results

### Socio demographic and economic characteristics

In this study, a total of 2544 women age 15–19 years who gave birth in the 5 years preceding the surveys or pregnant during the surveys were included. About three-fourth of study participants were 18 and 19 years of age 1921 (75.5%) and the overall mean age was 18 years. Majority of respondents, 1547 (60.8%) had no formal education and 1148 (45.3%) of them were orthodox Christians. Around 1453 (60.7%) of participants were unemployed and 1395 (57.6%) of women were from poor household wealth status. Most, 2284 (89.8%) of women were married while only 1106 (43.5%) of them had media exposure (Table 2).

### Reproductive history of participants

About two-third 1719 (67.6%) had one live child and 718 (28.2%) had two or more children. Eight hundred ninety one (66.0%) were fifteen year old or younger during their first sex. Eight hundred twenty two (38.5%) of respondents had a history of at least one Antenatal care (ANC) visit, and only 278 (13.0%) delivered their index child in a health facility. Most, 2283 (89.8%) knew at least one family planning method while 342 (13.4%) correctly knew that a women is most likely to conceive half way between two periods and only 587 (23.1%) of them had ever used contraceptive. Moreover, 1677 (66.0%) had wanted (planned) the then pregnancies, 657 (25.9%) of them had wanted later (miss-timed) pregnancies while 206 (8.1%) did not want their pregnancies. On the other hand, 132 (5.2%) of respondents had abortion (Table 3).

### Trends in unintended pregnancy among adolescents in Ethiopia

The magnitude of an unintended pregnancy among Ethiopian adolescents aged 15–19 years significantly decreased from 307 (41.4%) (95% CI: 35.7, 47.2%) in 2000 to 120 (25.1%) (95% CI: 18.9, 31.4%, p<0.001) in 2016.

Between the study period 2000 to 2016, the largest significant reduction in unintended pregnancy was observed among women with primary education (Diff = − 32.3; 95% CI: −48.4, −16.4), followed by those reside in metropolis regions (Diff = −29.0; 95% CI: −49.8, −8.3), those who had ever used contraceptive (Diff = −27.1; 95% CI: −43.4, −10.7) and whose age at first sex older than 15 years (Diff = −26.3; 95% CI: −44.3, −8.4). The magnitude of unintended pregnancy was also significantly reduced from 2000 to 2016 among women lived in Harari (Diff = −25.8; 95% CI: −46.7, −4.9), Tigray (Diff = −19.9; 95% CI: −36.5, −3.3) and Oromia (Diff = −13.6; 95% CI: −27.1, −0.2) regions. It was also significantly reduced among rural residents (Diff = −17.3; 95% CI: −26.3, −8.3), married women (Diff = −19.0; 95% CI: −27.6, −10.4), unemployed women (Diff = −23.5; 95% CI: −35.4, −11.5), women from poor household wealth (Diff = −20.0; 95% CI: −31.0,-9.0), women from male headed households (Diff = −20.9; 95%

**Table 2. Socio demographic and economic characteristics of adolescents in Ethiopia, 2000–2016.**

| Socio demographic and economic variables | | Number | Percent |
|---|---|---|---|
| Age | 15–17 | 623 | 24.5 |
| | 18–19 | 1921 | 75.5 |
| Residence | Urban | 228 | 8.9 |
| | Rural | 2316 | 91.1 |
| Educational status | No education | 1547 | 60.8 |
| | Primary | 871 | 34.2 |
| | Secondary and higher | 125 | 4.9 |
| Marital status | Not married | 260 | 10.2 |
| | Married | 2284 | 89.8 |
| Religion | Orthodox | 1148 | 45.3 |
| | Protestant | 389 | 15.3 |
| | Muslim | 946 | 37.3 |
| | Others | 54 | 2.1 |
| Employment status | Not working | 1453 | 60.7 |
| | Working | 940 | 39.3 |
| Family size | 2–3 | 1079 | 42.4 |
| | 4–5 | 846 | 33.3 |
| | > = 6 | 619 | 24.3 |
| Household wealth | Poor | 1395 | 57.6 |
| | Middle | 694 | 28.6 |
| | Rich | 335 | 13.8 |
| Regions category | Large central | 2344 | 92.2 |
| | Small peripheral | 150 | 5.9 |
| | Metropolis | 50 | 1.9 |
| Regions | Tigray | 177 | 6.9 |
| | Afar | 31 | 1.2 |
| | Amhara | 678 | 26.7 |
| | Oromia | 1176 | 46.2 |
| | Somali | 74 | 2.9 |
| | Benshangul-Gumuz | 34 | 1.4 |
| | SNNPR | 313 | 12.3 |
| | Gambella | 11 | 0.4 |
| | Harari | 8 | 0.3 |
| | Addis Ababa | 33 | 1.3 |
| | Dire Dawa | 9 | 0.4 |
| Media exposure | No | 1438 | 56.5 |
| | Yes | 1106 | 43.5 |
| Distance to health facility | Big problem | 1283 | 71.2 |
| | Not a big problem | 521 | 28.8 |
| Decision for health care | Not involved | 1413 | 55.6 |
| | Involved | 1131 | 44.4 |

CI: −29.8, −12.1), women who had media exposure (Diff = −19.1; 95% CI: −34.3, −3.9) (Table 4).

In addition, significant decline of unintended pregnancy was detected among women with home delivery (Diff = −18.0; 95% CI: −29.7, −6.3), women who had no abortion (Diff = −18.4; 95% CI: −27.1, −9.7) and those who had ANC history (Diff = −23.9; 95% CI: −36.8, −11.0).

**Table 3. Reproductive history of adolescents in Ethiopia, 2000–2016.**

| Variables | | Number | Percent |
|---|---|---|---|
| Parity | 0 | 107 | 4.2 |
| | 1 | 1719 | 67.6 |
| | 2 | 609 | 23.9 |
| | 3–4 | 109 | 4.3 |
| ANC visit history | No | 1312 | 61.5 |
| | Yes | 822 | 38.5 |
| Place of delivery | Home | 1865 | 87.0 |
| | Health facility | 278 | 13.0 |
| Mode of delivery | Non- Caesarean | 2116 | 98.8 |
| | Caesarean Section | 26 | 1.2 |
| Birth order | 1st | 2027 | 79.7 |
| | 2nd | 447 | 17.6 |
| | 3rd or 4th | 70 | 2.7 |
| Pregnancy plan status | Wanted then | 1677 | 66.0 |
| | Wanted later | 657 | 25.9 |
| | Not wanted at all | 206 | 8.1 |
| Knows any family planning | No | 260 | 10.2 |
| | Yes | 2283 | 89.8 |
| Ever use contraceptive | No | 1957 | 76.9 |
| | Yes | 587 | 23.1 |
| Knows ovulation period | Yes | 342 | 13.4 |
| | No | 2201 | 86.6 |
| Age at first sex(years) | $\leq$ 15 | 891 | 66.0 |
| | >15 | 460 | 34.0 |
| Abortion experience | No | 2412 | 94.8 |
| | Yes | 132 | 5.2 |

ANC: Antenatal care.

Similarly, it was significantly reduced among women from 2 to 3 family size (Diff = −15.4; 95% CI: −28.7, −2.2), from 4–5 family size (Diff = −18.5; 95% CI: −34.9, −2.1), women who had one live child (Diff = −14.9; 95% CI: −24.8, −4.9) and women who had 2–4 live children (Diff = −22.9; 95% CI: −41.2, −4.7) (Table 5).

### Trends of abortion among adolescent women in Ethiopia

The prevalence of abortion significantly decreased from 62 (8.3%) (95% CI: 5.2, 11.4%) in 2000 to 20 (4.1%) (95% CI: 1.3, 6.9%, p = 0.004) in 2016. Significant reduction in percentage point of abortion was observed among women with intended pregnancy (Diff = −7.5; 95% CI: −12.9, −2.1), women who had no media exposure (Diff = −6.7; 95% CI: −12.3, −1.2), women who had one live child (Diff = −6.1; 95% CI:−10.9−1.2) and women resided in large central regions (Diff = −4.5; 95% CI: −9.1,−0.1) (Table 6).

### Decomposition analysis for unintended pregnancy and abortion

After controlling the role of changes in compositional characteristics, age (Coeff = -0.41, 95% CI, -0.64, -0.18, p<0.001) and living in Somali regional state (Coeff = -2.21, 95%CI, -3.27, -1.15, p<0.001) showed a significance association with decline in unintended pregnancy

**Table 4. Trend of unintended pregnancy by socio demographic and economic variables among adolescent women in Ethiopia, 2000–2016.**

| Variables | 2000 %(95%CI) | 2005 %(95%CI) | 2011 %(95%CI) | 2016 %(95%CI) | 2000–2016 n(%) | 2016–2000 aDiff (95% CI) |
|---|---|---|---|---|---|---|
| **Age(years)** | | | | | | |
| 15–17 | 55.0(44.4–65.6) | 45.7(33.8–57.6) | 37.3(24.8-49-8) | 34.1(21.8–46.4) | 276(44.3) | -20.9(-37.2,-4.6) |
| 18–19 | 36.9(30.5–43.3) | 129.9(23.5–36.4) | 30.7(24.0–37.3) | 21.6(14.2–28.9) | 588(30.6) | -15.3(-25.1,-5.6)* |
| **Residence** | | | | | | |
| Urban | 43.6(27.8–59.4) | 40.8(23.5–58.1) | 23.9(9.1–38.8) | 38.7(15.9–61.5) | 86(37.6) | -4.9(-32.6, 22.8) |
| Rural | 41.1(34.9–47.4) | 33.4(27.1–39.7) | 32.7(26.0–39.8) | 23.8(17.4–30.3) | 778(33.6) | -17.3(-26.3,-8.3)* |
| **Education** | | | | | | |
| No education | 36.6(30.2–43.1) | 31.9(25.1–38.8) | 25.4(17.3–33.5) | 16.6(5.6–27.6) | 479(31.0) | -20.0(-32.8,-7.2)* |
| Primary | 60.0(46.4–73.8) | 40.3(28.6–51.9) | 39.1(30.5–47.6) | 27.7(19.4–36.0) | 330(37.9) | -32.3(, -48.4,-16.4)* |
| Secondary & higher | 58.2(37.8–78.5) | 27.8(0.9–54.7) | 25.9(8.9–43.0) | 40.0(18.1–62.0) | 54(43.4) | -18.2(-48.1, 11.8) |
| **Marital status** | | | | | | |
| Not married | 55.9(40.8–76.1) | 66.9(53.3–80.6) | 61.7(39.7–83.7) | 47.0(24.6–69.4) | 148(57.2) | -8.9(-35.9,18.1) |
| Married | 40.0(33.9–46.0) | 30.9(24.8–37.0) | 28,9(22.1–35.6) | 21.0(15.0–27.1) | 715(31.5) | -19.0(-27.6,-10.4)* |
| **Employment** | | | | | | |
| Not employed | 43.8(34.4–53.2) | 32.0(25.3–38.7) | 29.7(21.8–37.6) | 20.3(13.0–27.6) | 454(31.3) | -23.5(-35.4,-11.5)* |
| Employed | 39.7(32.0–47.5) | 41.3(30.5–52.1) | 34.8(26.4–43.2) | 32.3(20.7–43.8) | 348(37.0) | -7.4(-21.4,6.4) |
| **Household wealth** | | | | | | |
| Poor | 41.1(34.5–47.6) | 31.6(24.5–38.7) | 26.4(17.5–35.2) | 21.1(12.3–29.8) | 443(31.8) | -20.0(-31.0,-9.0)* |
| Middle | 43.1(31.9–54.3) | 34.4(24.6–44.1) | 37.5(23.3–51.8) | 27.8(16.6–39.1) | 251(36.3) | -15.3(-31.2,0.7) |
| Rich | 35.3(8.3–62.2) | 2.4(0.7–5.4) | 37.4(27.8–46.9) | 30.8(17.6–44.1) | 112(33.5) | -4.5(-34.5, 25.7) |
| **Regions** | | | | | | |
| Tigray | 32.7(19.8–45.5) | 8.2(1.8–14.6) | 29.5(16.9–41.9) | 12.8(2.3–23.2) | 40(22.5) | -19.9(-36.5, -3.3) |
| Afar | 21.3(9.7–32.8) | 24.5(10.5–38.6) | 13.0(2.2–23.8) | 19.7(10.7–28.7) | 6(20.1) | -1.6(-15.7, 12.7) |
| Amhara | 50.3(40.7–59.8) | 33.0(22.9–43.1) | 35.5(20.8–50.2) | 40.4(20.6–60.2) | 277(40.9) | -9.9(-31.9.1,12.1) |
| Oromia | 37.7(27.9–47.5) | 38.5(14.8–33.4) | 32.4(22.8–42.1) | 24.1(14.8–33.4) | 395(33.7) | -13.6(-27.1, -0.2)* |
| Somali | 1.5(0.9–4.8) | 6.1(2.7–14.9) | 8.3(0.8–17.3) | 0(0) | 3(4.2) | -1.5(-4.8, 1.7) |
| Benshangul-Gumuz | 35.6(22.2–48.9) | 20.9(6.0–35.8) | 31.0(21.3–40.7) | 29.7(12.0–47.5) | 10(29.9) | -5.9(-28.0,16.4) |
| SNNPR | 38.0(20.7–55.3) | 41.2(28.5–53.9) | 29.1(13.9–44.2) | 28.1(14.9–41.2) | 108(34.4) | -9.9(-31.7, 11.8) |
| Gambella | 40.0(21.6–58.3) | 23.5(10.2–36.7) | 43.3(27.2–59.4) | 29.3(11.6–47.0) | 4(35.4) | -10.7(-36.2,14.9) |
| Harari | 33.8(14.4–53.1) | 22.5(13.2–31.9) | 39.1(19.9–58.4) | 8.0(0.1–15.9) | 2(25.8) | -25.8(-46.7,-4.9)* |
| Addis Ababa | 60.5(39.4–81.5) | 30.8(10.7–50.8) | 67.3(41.6–92.9) | 31.9(12.2–51.6) | 16(47.1) | -28.6(-57.4, 0.3) |
| Dire Dawa | 41.1(20.4–62.2) | 11.1(0.3, 22.5) | 30.6(11.3–49.9) | 16.7(3.3–30.2) | 2(23.5) | -24.4(-49.5, 0.3) |
| **Family size** | | | | | | |
| 2–3 | 40.8(32.4–49.3) | 30.4(22.4–38.4) | 30.2(21.5–38.9) | 25.4(15.2–35.6) | 340(32.1) | -15.4(-28.7, -2.2)* |
| 4–5 | 44.7(35.2–54.1) | 34.3(24.2–44.3) | 23.5(14.2–32.7) | 26.2(12.7–39.7) | 283(33.4) | -18.5(-34.9, -2.1)* |
| >5 | 37.7(26.0–49.5) | 40.9(28.3–53.5) | 47.0(33.8–60.2) | 23.9(14.5–33.4) | 231(37.3) | -13.8(-28.9, 1.3) |
| Media exposure | | | | | | |
| No | 36.5(30.0–42.9) | 32.8(25.9–39.5) | 28.8(17.6–40.0) | 21.1(13.3–29.0) | 444(30.9) | -15.4(-25.5,-5.2)* |
| Yes | 51.0(40.8–61.2) | 35.6(25.0–46.3) | 33.8(26.5–41.2) | 31.9(20.7–43.1) | 419(37.9) | -19.1(-34.3,-3.9) |
| Overall | 41.4(35.7–47.2) | 33.9(27.9–39.9) | 31.9(25.7–38.3) | 25.1(18.9–31.4) | 863(34.0) | -16.3 (-24.8, -7.8)* |

*CI: Confidence Interval, aDiff: Percentage change in prevalence of unintended pregnancy between 2000 to 2016, SNNPR: South Nations Nationalities and Peoples Region, *: Significant for trend at p-value <0.05.

whereas; living in Benshangul-Gumuz regional state (Coeff = -0.17, 95%CI, -0.32, -0.19, p = 0.03) showed significance association with decline in abortion from socio demographic and economic variables (Table 7).

**Table 5. Trend of unintended pregnancy by reproductive variables among adolescent women in Ethiopia, EDHS 2000–2016.**

| Variables | 2000 n (%) | 2005 n (%) | 2011 n (%) | 2016 n (%) | 2000–2016 n(%) | 2016–2000 aDiff (95% CI) |
|---|---|---|---|---|---|---|
| Parity | | | | | | |
| 0 | 31.7(12.6–50.7) | 8.4(6.7–23.5) | 53.4(11.1–95.8) | 38.5(8.5–85.5) | 31(28.9) | 6.8(-4.4, 57.8) |
| 1 | 38.8(31.9–45.7) | 36.0(28.9–43.1) | 29.4(23.2–35.6) | 24.0(16.8–31.1) | 562(32.7) | -14.9(-24.8, -4.9)* |
| 2–4 | 50.0(39.3–60.7) | 33.7(21.5–45.9) | 35.9(23.8–47.9) | 27.1(12.3–41.8) | 270(37.7) | -22.9(-41.2, -4.7)* |
| Sex of HH Head | | | | | | |
| Male | 41.6(35.7–47.5) | 33.6(27.3–40.0) | 31.6(24.5–38.8) | 20.6(14.1–27.2) | 729(33.3) | -20.9(-29.8,-12.1)* |
| Female | 40.0(24.2–55.8) | 36.3(22.4–50.1) | 33.6(21.0–46.1) | 45.4(31.6–59.2) | 134(38.5) | 5.4(-15.5, 26.4) |
| Place of delivery | | | | | | |
| Home | 44.4(37.7–51.1) | 33.6(26.9–40.3) | 35.2(27.4–43.1) | 26.4(16.9–35.9) | 676(36.3) | -18.0(-29.7,-6.3)* |
| Health facility | 32.0(13.9–50.2) | 43.3(23.0–63.5) | 18.1(11.7–35.1) | 24.9(14.5–35.5) | 75(26.9) | -7.1(-28.0, 13.9) |
| Ever use FP | | | | | | |
| No | 39.4(33.2–45.5) | 34.8(28.7–40.9) | 31.5(23.5–39.6) | 24.8(17.2–32.3) | 666(34.1) | -14.6(-24.3, -4.9)* |
| Yes | 52.7(39.5–65.8) | 28.1(14.0–42.2) | 33.0(24.3–41.7) | 25.6(15.9–35.4) | 197(33.6) | -27.1(-43.4,-10.7)* |
| Ovulation period | | | | | | |
| Knows | 44.6(29.7–59.5) | 32.3(17.7–46.9) | 26.6(10.9–42.4) | 31.1(16.1–45.9) | 114(33.5) | -13.5(-34.7,7.5) |
| Don't know | 41.0(34.9–47.2) | 34.1(27.5–40.8) | 32.9(26.6–39.4) | 23.8(16.7–30.8) | 749(34.1) | -17.2(-26.7,-7.8)* |
| Age at first sex | | | | | | |
| < = 15 | 46.5(36.2–56.9) | 38.2(29.9–46.5) | 43.4(28.7–58.1) | 28.9(20.1–37.7) | 339(38.1) | -17.6(-31.3, -4.0) |
| >15 | 46.2(30.1–61.4) | 37.8(24.1–51.4) | 41.8(20.9–62.7) | 19.9(10.2–29.5) | 147(32.0) | -26.3(-44.3, -8.4)* |
| Abortion | | | | | | |
| No | 43.5(37.5–49.5) | 35.0(28.9–41.1) | 32.9(26.5–39.3) | 25.1(18.9–31.3) | 843(34.9) | -18.4(-27.1,-9.7)* |
| Yes | 18.7(5.7–31.8) | 0(0) | 13.4(3.1–29.9) | 26.1(12.9–65.1) | 20(15.5) | 7.4(-33.9,48.5) |
| ANC history | | | | | | |
| No | 41.8(34.3–49.4) | 35.8(28.8–42.9) | 34.2(25.2–43.1) | 31.3(15.0–47.7) | 487(37.1) | -10.5(-285,7.6) |
| Yes | 47.9(37.1–58.7) | 28.6(16.5–40.6) | 32.9(22.5–43.3) | 24.0(17.1–30.9) | 262(21.9) | -23.9(-36.8,-11.0)* |

*CI: Confidence Interval, aDiff: Percentage change in prevalence of unintended pregnancy between 2000 to 2016, HH: House Hold, FP: Family Planning, ANC: Ante Natal Care, *: Significant for trend at p-value <0.05.

From a total of nine reproductive history related variables entered into a multilevel mixed effect decomposition logistic regression model, an exposure to media (Coeff = -0.60, 95%CI, -0.87, -0.33, *p*<0.001) showed a significance association with decline in unintended pregnancy. Similarly, from eight reproductive health related variables entered into final multilevel mixed effect decomposition logistic regression model; ANC history (Coeff = -0.81, 95%CI, -1.45, -0.17, *p* = 0.01) was significantly associated with induced abortion decline from 2000–2016 (Table 8).

## Discussion

We assessed the determining factors for trends of unintended pregnancy and abortion among adolescent women in Ethiopia for the years and we found that there was a significant overall decline. Although, this finding was in line with similar other studies [4,15–19], unintended pregnancy rate in Ethiopia still remains high. The significant decline of unintended pregnancy may be due to the impacts of the national reproductive health interventions that were effectively implemented in Ethiopia [12–15]. Factors associated with significant decline of unintended pregnancy during the past decades included; adolescent women being aged above 17 years and adolescent women who used to live in Somali regional state.

**Table 6. Trends of induced abortion by variables among adolescent women in Ethiopia, EDHS 2000–2016.**

| Variables | 2000 %(95%CI) | 2005 %(95%CI) | 2011 %(95%CI) | 2016 %(95%CI) | 2000–2016 n(%) | 2016–2000 aDiff% (95% CI) |
|---|---|---|---|---|---|---|
| **Age(years)** | | | | | | |
| 15–17 | 4.6(0.4–8.8) | 2.0(0.4–4.4) | 3.9(1.4–9.2) | 3.1(1.6–7.8) | 21(3.4) | -1.5(-7.8,4.8) |
| 18–19 | 9.5(5.7–13.3) | 3.6(0.6–6.6) | 4.7(1.9–7.5) | 4.4(0.8–8.1) | 111(5.7) | -5.0(-10.3,0.3) |
| **Residence** | | | | | | |
| Urban | 5.9(1.5–13.3) | 11.2(0.7–21.7) | 11.1(1.4–23.7) | 1.2(0.7–3.1) | 17(7.4) | -4.7(-12.4,3.1) |
| Rural | 8.6(5.2–11.9) | 2.6(0.2–4.9) | 3.9(1.5–6.4) | 4.3(1.3–7.4) | 115(4.9) | -4.2(-8.8(0.3) |
| **Education** | | | | | | |
| No education | 8.1(5.2,12.3) | 4.0(1.8,8.7) | 4.3(2.1,8.8) | 6.6(2.1,19.5] | 91(5.8) | -1.4(-9.6,6.8) |
| Primary and above | 8.0(4.2,18.4) | 1.1(0.3,3.5) | 4.7(2.0,10.8) | 2.9(1.2,6.9) | 41(4.1) | -6.1(-13.3,1.1) |
| **Employment** | | | | | | |
| Not employed | 8.3(4.3,15.6) | 3.2(1.3,7.5) | 4.6(2.1,9.8] | 3.3(1.3,8.3) | 65(4.5) | -5.0(-11.3,1.3) |
| Employed | 8.5(4.7,14.9) | 3.3(1.1,9.9) | 3.9(1.7,8.0) | 5.3(1.8,14.4) | 54(5.7) | -3.3(-10.7,4.1) |
| **Household wealth** | | | | | | |
| Poor | 9.9(6.5,14.6) | 4.3(1.6,10.6) | 3.8(1.8,7.9) | 6.4(1.4,11.4) | 89(6.4) | -3.4(-9.8,2.9) |
| Middle | 2.6[0.5,13.4) | 1.1(0.2,5.2) | 5.7(1.8,16.9) | 2.6(1.3,6.5) | 19(2.7) | -0.1(-5.9,5.9) |
| Rich | 14.1(2.8,48.7) | 29.4(10,61.0) | 5.0(1.9,13.2) | 0.6(0.4–1.6) | 17(4.9) | -13.5(-34.8,7.7) |
| **Regions category** | | | | | | |
| Large central | 8.5(5.7,12.4) | 3.0(1.3,6.8) | 4.4[2.4,7.9] | 4.0(1.8,8.6) | 121(5.2) | -4.5(-9.1,-0.1) |
| Small peripheral | 4.1(1.7,9.7) | 4.1[0.5,25.1] | 3.6(0.7,15.4) | 5.8(2.7,12.2) | 7(4.4) | 1.7(-3.9,7.3) |
| Metropolis | 6.3(2.1,17.4) | 8.7(2.8,23.6) | 18.4(4.3,53.2) | 2.2(0.8,6.5) | 4(8.2) | -4.1(-11.1,3.0) |
| **Family size** | | | | | | |
| 2–3 | 11.5(6.3–16.7) | 4.3(1.0–7.7) | 4.8(1.0–8.6) | 5.9(0.4–11.9) | 73(6.9) | -5.5(-13.4,2.4) |
| 4–5 | 5.2(0.8–9.7) | 1.0(0.8–9.7) | 3.5(0.1–7.1) | 3.9(1.5–9.3) | 29(3.3) | -1.4(-8.4,5.6) |
| >5 | 6.8(0.6–14.2) | 4.8(3.7–13.2) | 5.6(0.4–11.6) | 1.7(1.0–4.4) | 30(4.8) | -5.1(-12.9,2.8) |
| **Parity** | | | | | | |
| 0 | 10.7(3.3–24.8) | 17.6(3.1–38.1) | 3.2(0.4–9.9) | 0.8(0.1–2.7) | 11(10.2) | -9.9(-24.1,4.2) |
| 1 | 9.7(5.9–13.6) | 2.4(0.6–4.2) | 6.1(2.7–9.4) | 3.6(0.6–6.7) | 98(5.7) | -6.1(-10.9,-1.2)* |
| 2–4 | 4.1(1.5–9.7) | 2.7(2.4–7.7) | 1.1(0.4–2.5) | 5.9(2.8–14.5) | 23(3.2) | 1.8(-8.5,12.1) |
| **Sex of HH Head** | | | | | | |
| Male | 8.6(5.2–11.9) | 3.5(0.9–6.1) | 5.6(2.5–8.7) | 4.9(1.4–8.4) | 127(5.8) | -3.6(-8.5,1.2) |
| Female | 5.7(1.8–13.2) | 0.7(0.6–1.9) | 0.01(0.02–0.07) | 0.2(0.1–0.6) | 5(1.4) | -5.5(-12.9,2.1) |
| **Media exposure** | | | | | | |
| No | 10.2(6.0–14.4) | 3.4(0.1–6.9) | 5.8(0.9–10.8) | 3.5(0.1–7.1) | 88(6.1) | -6.7(-12.3,-1.2)* |
| Yes | 4.5(0.5–8.5) | 2.8(0.3–5.4) | 3.8(1.2–6.5) | 5.1(0.2–10.3) | 44(3.9) | 0.6(-5.9,7.2) |
| **Place of delivery** | | | | | | |
| Home | 7.9(4.4–11.5) | 3.3(0.5–6.1) | 3.4(0.9–5.9) | 5.0(0.3–9.7) | 93(4.9) | -2.9(-8.8,2.9) |
| Health facility | 18.2(0.3–36.2) | 11.2(2.0–24.3) | 11.2(3.8–26.3) | 1.8(0.9–4.5) | 21(7.4) | -16.5(-34.7,1.7) |
| **Ever use FP** | | | | | | |
| No | 8.6(5.2–12.0) | 3.0(0.4–5.6) | 4.5(1.5–7.5) | 3.9(0.7–7.2) | 103(5.3) | -4.6(-9.4,0.1) |
| Yes | 6.5(0.4–13.4) | 4.4(0.6–9.3) | 4.7(0.2–9.2) | 4.2(1.1–9.5) | 29(4.8) | -2.3(-11.0,6.4) |
| **Pregnancy plan** | | | | | | |
| Intended | 11.5(6.9–16.1) | 4.9(1.4–8.3) | 5.8(2.4–9.2) | 4.0(1.2–6.8) | 111(6.6) | -7.5(-12.9,-2.1)* |
| Unintended | 3.7(0.9–6.6) | 0 | 1.9(1.0.5–4.3) | 4.2(3.4–11.9) | 21(2.4) | 0.5(-7.7,8.7) |
| **ANC history** | | | | | | |
| No | 8.1(3.9–12.2) | 4.6(1.0–8.2)) | 3.2(0.7–5.7) | 5.4(3.7–14.5) | 72(5.5) | -2.7(-12.7,7.3) |
| Yes | 10.9(3.3–18.6) | 1.0(0.5–2.4) | 5.7(0.4–10.9) | 3.2(0.4–6.0) | 41(4.9) | -7.7(-15.9,0.4) |

*(Continued)*

**Table 6.** (Continued)

| Variables | 2000 %(95%CI) | 2005 %(95%CI) | 2011 %(95%CI) | 2016 %(95%CI) | 2000–2016 n(%) | 2016–2000 aDiff% (95% CI) |
|---|---|---|---|---|---|---|
| Overall | 8.3(5.2–11.4) | 3.2(0.9–5.5) | 4.6(2.1–7.0) | 4.1(1.3–6.9) | 5.2(3.9–6.7) | -4.2 (-8.4–0.03)* |

*CI: Confidence Interval, aDiff: Percentage change in prevalence of unintended pregnancy between 2000 to 2016, HH: House Hold, FP: Family Planning, ANC: Ante Natal Care, *: Significant for trend at p <0.05.

Similarly, the trend of induced abortion significantly declined from 8.3% in 2000 to 4.1% in 2016 in Ethiopia and still the magnitude of abortion is high. This figure is consistent when compared to similar other studies [6–8]. This significant reduction in an induced abortion may be attributed to the relatively increased maternal health service utilization like modern contraceptive use by the currently married women (35%)) in 2016 [5,12] compared to (6%) in 2000 [12,17,20].

The trend of unintended pregnancy significantly declined among adolescent women aged 18 years and above compared to adolescent women aged below 18 years. Studies conducted in developing countries indicated the risk of unplanned pregnancies decreases with relatively older age [21–26]. This might be due to older women had relatively better knowledge on contraceptive methods to prevent unintended pregnancy and lower contraceptive failure rate [21,24]. Moreover, as they are getting older, women might also become more literate about the importance and accessibility of reproductive or maternal health services. In addition, this could be also as the results of older women are less likely to engage in risky sexual behaviors such as unprotected sexual intercourse and sex under the influence of drinking alcohol [25,26]. However, other literatures revealed a negative relationship between age and unintended pregnancy [23,27,28]. This finding might be related to the fact that adult women might already have the desirable number of children and considered any additional pregnancy as mistimed or unwanted.

The current finding also suggests that the trend of unintended pregnancy significantly reduced among adolescent women residing in Somali regional state of Ethiopia. Evidences indicated that in some Ethiopian regions including Afar, Benshangul-Gumuz, Somali, and Oromia; half or more of all girls are married before age 18 [29]. Child marriage is significantly associated with a history of rapid repeat childbirth and not using contraception before first childbirth; which leads to the pregnancy to be wanted [30]. Thus, the significant decrease in unintended pregnancy among adolescent women in Somali region may be due to the increased child marriage that may end with intended pregnancy [29,30–34]. Furthermore, Somali and Afar regions have the lowest proportion of women and men with desire to limit childbearing and have the highest total wanted fertility rate in Ethiopia compared to the other regions [32]. The lowest average of contraceptive use rate (8.8%) in Ethiopia in 2021 was recorded in Somali region [33].

Induced abortion significantly declined among adolescent women with ANC history. This finding is consistent with studies conducted in Kenya [25], in Ethiopia [35–39], in South Africa [40] and in United States [41]; reported that previous exposure to ANC had influenced induced abortion. This might be explained as women having a previous exposure to ANC follow-up and have given birth might be influenced to have an additional baby rather than planning for abortion [24–27].

Similarly, induced abortion significantly declined among adolescent women residing in Benshangul-Gumuz regional state of Ethiopia during the study period. This may be explained from different angles. The abortion rate continued to be lowest (6.7 per 1,000 in the least densely populated and most traditional rural regions (Afar, Benshangul-Gumuz, Gambella and Somali),

**Table 7. Decomposition analysis for trends of unintended pregnancy and abortion by socio demographic and economic variables among adolescent women in Ethiopia, 2000–2016.**

| Variables | Unintended pregnancy | | | Abortion | | |
|---|---|---|---|---|---|---|
| | Coeff. | P-value | 95%CI | Coeff. | P-value | 95%CI |
| **Age(years)** | | | | | | |
| 15–17 | Ref | | | Ref | | |
| 18–19 | -0.41 | <0.001 | -0.64, -0.18 | 0.41 | 0.12 | -0.11, 0.34 |
| **Residence** | | | | | | |
| Urban | Ref | | | Ref | | |
| Rural | -0.42 | 0.09 | -0.69, 0.05 | 0.81 | 0.025 | 0.10, 1.51 |
| **Education** | | | | | | |
| No education | Ref | | | Ref | | |
| Primary | 0.31 | 0.005 | 0.09, 0.53 | -0.10 | 0.67 | -0.56, 0.36 |
| Secondary & higher | -0.03 | 0.89 | -0.47, 0.41 | -0.22 | 0.62 | -1.10, 0.66 |
| **Marital status** | | | | | | |
| Not married | Ref | | | | | |
| Married | 1.08 | <0.001 | 0.70. 1.70 | -0.33 | 0.49 | -1.32, 0.62 |
| **Employment** | | | | | | |
| Not employed | Ref | | | Ref | | |
| Employed | -0.05 | 0.82 | -0.48, 0.37.8 | -0.22 | 0.64 | -1.18, 0.73 |
| **Household Wealth** | | | | | | |
| Poor | Ref | | | Ref | | |
| Middle | 0.12 | 0.49 | -0.22, 0.48.0 | 0.61 | 0.12 | -1.14, 1.36 |
| Rich | 0.22 | 0.20 | -0.11, 0.56 | 0.22 | 0.19 | -0.11, 0.56 |
| **Regions** | | | | | | |
| Tigray | 0.26 | <0.001 | -0.21, 0.74 | -0.03 | 0.96 | -0.93, 0.88 |
| Afar | 0.01 | 0.96 | -0.62, 0.66 | 0.73 | 0.24 | -1.11, 1.21 |
| Amhara | 1.06 | <0.001 | 0.63, 1.52 | 0.32 | 0.42 | -0.46, 1.11 |
| Oromia | 0.8 | 0.23 | 0.32, 0.94 | 0.19 | 0.64 | -0.60, 0.98 |
| Somali | -2.21 | <0.001 | -3.27, -1.15 | 0.23 | 0.60 | -0.64, 1.10 |
| Benshangul-Gumuz | 0.59 | 0.014 | 0.12, 1.27 | -0.17 | 0.03 | -0.32, -0.19 |
| SNNPR | 0.86 | 0.001 | 0.37, 1.36 | -0.25 | 0.64 | -1.30, 0.81 |
| Gambella | 0.74 | 0.03 | 0.25, 1.25 | 0.11 | 0.84 | -0.89, 1.11 |
| Harari | 0.33 | 0.21 | -0.19, 0.87 | 0.45 | 0.34 | -0.47, 1.38 |
| Dire Dawa | 0.02 | 0.96 | -0.62, 0.66 | 0.05 | 0.93 | -1.11, 1.21 |
| Addis Ababa | Ref | | | Ref | | |
| **Family size** | | | | | | |
| 2–3 | Ref | | | Ref | | |
| 4–5 | -0.18 | 0.18 | -0.45, 0.08 | 0.52 | 0.07 | -0.04, 1.08 |
| >5 | 0.03 | 0.86 | -0.25, 0.29 | -0.14 | 0.66 | -0.78, 0.49 |

*CI: Confidence Interval, SNNPR: South Nations Nationalities and Peoples Region, HH: House Hold, FP: Family Planning, ANC: Ante Natal Care.

apparently for the reason that of limited access to services, lower utilization of abortion services or both [42]. Moreover, abortion in Somalia is legal only to save the pregnant person's life [8].

## Strengths and limitations of the study

The research is based on nationally representative data with large sample size so that the observed findings more likely to show the trends of unintended pregnancy and abortion

**Table 8. Decomposition analysis for trends of unintended pregnancy and abortion by reproductive history variables among adolescent women in Ethiopia, 2000–2016.**

| Variables | Unintended Pregnancy | | | Abortion | | |
|---|---|---|---|---|---|---|
| | Coeff. | p-value | 95%CI | Coeff. | p-value | 95%CI |
| Parity | | | | | | |
| 0 | Ref | | | Ref | | |
| 1 | 0.37 | 0.24 | -0.23, 0.98 | 0.04 | 0.96 | -1.43, 1.51 |
| 2–4 | 0.35 | 0.25 | -0.28, 0.98 | 0.26 | 0.73 | -1.24, 1.77 |
| Sex of HH Head | | | | | | |
| Male | Ref | | | Ref | | |
| Female | 0.04 | 0.82 | -0.27, 0.34 | 0.93 | 0.05 | -0.01, 1.88 |
| Media exposure | | | | | | |
| No | Ref | | | | | |
| Yes | -0.60 | <0.001 | -0.87, -0.33 | 0.36 | 0.29 | -0.30, 1.01 |
| Place of delivery | | | | | | |
| Home | Ref | | | Ref | | |
| Health facility | 0.35 | 0.043 | 0.007, 0.71 | 0.58 | 0.16 | -0.23, 1.40 |
| Ever use FP | | | | | | |
| No | Ref | | | Ref | | |
| Yes | 0.21 | 0.19 | -0.10, 0.53 | 0.06 | 0.88 | -0.66, 0.77 |
| Ovulation period | | | | | | |
| Knows | -0.11 | 0.55 | -0.47, 0.25 | 0.18 | 0.70 | -0.71, 1.07 |
| Don't know | Ref | | | Ref | | |
| Age at first sex | | | | | | |
| < = 15 | Ref | | | | | |
| >15 | -0.04 | 0.76 | -0.33, 0.24 | 0.31 | 0.39 | -0.39, 1.02 |
| Abortion | | | | | | |
| No | Ref | | | | | |
| Yes | 0.76 | 0.047 | -1.50, -0.01 | | | |
| ANC history | | | | | | |
| No | Ref | | | Ref | | |
| Yes | 0.18 | 0.23 | -0.11, 0.47 | -0.81 | 0.01 | -1.45, -0.17 |

*CI: Confidence Interval, HH: House Hold, FP: Family Planning, ANC: Ante Natal Care.

among adolescent women in Ethiopia. It has the potential to provide evidence for policy-makers and program planners to design appropriate intervention strategies both at national and regional levels because the estimates are based on the national survey data. Nevertheless, the study had limitations because of the Ethiopian demographic and health surveys data are mostly based on respondents' self-report and could have the possibility of recall bias. In addition, it is difficult to establish a cause-effect relationship between the outcomes and independent variables due to the research is based on cross-sectional data. Moreover, the precision of trend analysis by regions in Ethiopia may be questioned as the distribution of unintended pregnancy and induced abortion among adolescent women across some regions in Ethiopia is small and may give wrong impressions for readers. Hence, the findings need to be interpreted with caution.

## Conclusion

The trends for both unintended pregnancy and induced abortion significantly declined over the study period. Despite this reduction, unintended pregnancies and induced abortion are

high in Ethiopia. The significant decline in unintended pregnancy among adolescent women was observed among adolescent women aged older than 18 years, living in Somali regional state and exposure to media whereas; the significant decline in induced abortion is observed among adolescent women living in Benshangul-Gumuz regional state and having previous ANC service utilization history. Intervention strategies including expansion of an exposure to media to increase their knowledge and special attention to regions with high child marriage in Ethiopia might decrease unintended pregnancy and induced abortion among adolescent women. Moreover, scaling up of MCH services including ANC services can keep the momentum of decline in an induced abortion.

## Acknowledgments

The authors would like to acknowledge that the Ethiopian Demographic and Health Survey 2016 data used in this study were obtained from the DHS office; they have given permission to access the data, after we have prepared the proposal on the title.

## Author Contributions

**Conceptualization:** Tiruwork Amare, Fasil Tessema, Tamrat Shaweno.

**Data curation:** Tiruwork Amare, Fasil Tessema.

**Formal analysis:** Tiruwork Amare.

**Investigation:** Tiruwork Amare, Tamrat Shaweno.

**Methodology:** Tiruwork Amare, Fasil Tessema, Tamrat Shaweno.

**Project administration:** Tiruwork Amare.

**Resources:** Tiruwork Amare.

**Software:** Tiruwork Amare, Fasil Tessema, Tamrat Shaweno.

**Supervision:** Tiruwork Amare, Fasil Tessema, Tamrat Shaweno.

**Validation:** Tiruwork Amare, Fasil Tessema, Tamrat Shaweno.

**Visualization:** Tiruwork Amare, Fasil Tessema, Tamrat Shaweno.

**Writing – original draft:** Tiruwork Amare, Fasil Tessema, Tamrat Shaweno.

**Writing – review & editing:** Tiruwork Amare, Fasil Tessema, Tamrat Shaweno.

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
