## [Decision Letter · Decision Letter 0]

11 Oct 2023

PONE-D-23-09938Trend of Unintended Pregnancy, Induced Abortion and Associated Factors among Adolescents in Ethiopia: An evidence from Multilevel mixed-effects decomposition analysis of 2000 - 2016 EDHS DataPLOS ONE

Dear Dr. Shewano,

Thank you for submitting your manuscript to PLOS ONE. After careful consideration, we feel that it has merit but does not fully meet PLOS ONE’s publication criteria as it currently stands. Therefore, we invite you to submit a revised version of the manuscript that addresses the points raised during the review process.

We look forward to receiving your revised manuscript.

Kind regards,

Takele Gezahegn Demie, MPH

Academic Editor

PLOS ONE

Journal Requirements:

"Didn't receive any fund"

"None to declare"

Additional Editor Comments:

Dear Author,

Thank you for submitting your work to PLOS ONE for possible publication.

Reviewers suggested minor revisions and you have to address the reviewers' comments and resubmit your manuscript.

Best wishes!

Reviewers' comments:

Reviewer's Responses to Questions

**Comments to the Author**

1. Is the manuscript technically sound, and do the data support the conclusions?

Reviewer #1: Yes

Reviewer #2: Yes

Reviewer #3: Yes

2. Has the statistical analysis been performed appropriately and rigorously? 

Reviewer #1: Yes

Reviewer #2: Yes

Reviewer #3: Yes

3. Have the authors made all data underlying the findings in their manuscript fully available?

Reviewer #1: Yes

Reviewer #2: Yes

Reviewer #3: Yes

4. Is the manuscript presented in an intelligible fashion and written in standard English?

Reviewer #1: Yes

Reviewer #2: No

Reviewer #3: Yes

5. Review Comments to the Author

Reviewer #1: Title: Trend of Unintended Pregnancy, Induced Abortion and Associated Factors among Adolescents in Ethiopia: An evidence from multilevel mixed-effects decomposition analysis of 2000 - 2016 EDHS Data

General comments

The authors explored important insights that may contribute to the domain of knowledge in the trends of unintended pregnancy, induced abortion, and its associated factors in Ethiopia. This may contribute to the intervention in adolescent health services in the country. However, additional work is needed to be considered for publication in its current form. For instance, the language usage may need critical proofreading since there are many language errors. In the methodology sections, there are very important clarifications. The results also need clarification. Therefore, it should address those issues before it is considered for publication.

Specific Comments

Page 1-2: Abstract

• Background: 'Although the trends of unintended pregnancy and induced abortion have declined over time in Ethiopia’ if the trend is known. What is the necessity of the trend analysis?

• Methods: The age category of the study participants should be clearly presented in this section. Methods in which the data collected should be described to show clarity for the readers the analysis methods should also be clear at this point.

• Results: the reduction is better if it is presented in percent (with confidence intervals), and the direction of the association should be clear with reference categories.

Page 3-4: Introduction

• Paragraph 3, ‘Results from the EDHS 2016 report indicate that unintended pregnancy and induced abortion are still high.' What was the comparison—high or low?

• The introduction lacks an explanation of the relevancy of the study and the value it adds to the scientific community.

Page 4-8: Methods

• ‘According to the Ethiopian census report, by the year 2018, the estimated population of Ethiopia was 106.8 million; of this, 24.2% were adolescents (from 9 to 18 years old), and 11.7% were 15 to 19 years old. Adolescent girls were 5.8% of the total population.’ References are needed for this information.

• Table 1: What were the criteria to select those variables? For me, it is important to explain the justification for selecting those variables. Since EDHS has so many variables, it may need expert consultation or other selection methods. (e.g., data mining approach)

• Data management: it is not clear how the incomplete data issues were addressed in this analysis.

Page 8-19: Results and discussion

• The result and discussion section were well written. However, language proofreading is needed.

Reviewer #2: Firstly, I would like to appreciate the authors for their work. my concerns are as follows:

1. there is an editorial error in some parts of your document specially duplication of words (see the abstract/introduction and method section).

2. most of the data/figures presented in the introduction section are outdated and you better rewrite it with Up-to-date data

3.The discussion section is not well discussed. For example, you have mentioned that the prevalence of modern contraceptive usage in Ethiopia in 2016 as high and as this has special contribution for the reduction of induced abortion. how can you say this figure high though it is far less than half/50%?

3. the explanation given for Somali region is not convincing. You said that it could be due to NGO....is there specific NGO working in Somali region specifically working in this area?

4. The recommendation also lacks clarity and you better rewrite it.

Reviewer #3: In the conclusion section, the authors recommend interventions tailored to developing regions in Ethiopia. How can we say a significant association between a factors in two emerging regions implies a problem for the four developing regions in Ethiopia? On top of this, the authors should define developing regions in Ethiopia to inform the readers.

The conclusion is not different from the discussion section. I suggest to the authors that they revise the conclusion part.

It is better to rephrase repetitive words and phrases throughout the article. E.g., the title of the article is mentioned in the abstract and introduction. And also, in "this study," at the beginning of all paragraphs of the discussion and conclusion.

6. PLOS authors have the option to publish the peer review history of their article (what does this mean?). If published, this will include your full peer review and any attached files.

Reviewer #1: No

Reviewer #2: No

Reviewer #3: **Yes: **Mamo Dereje Alemu

---

## [Author Response · Author response to Decision Letter 0]

26 Dec 2023

Authors’ response/'Response to Reviewers' comments

Reviewer 1**************************************

Title: Trend of Unintended Pregnancy, Induced Abortion and Associated Factors among Adolescents in Ethiopia: An evidence from multilevel mixed-effects decomposition analysis of 2000 - 2016 EDHS Data

General comments

The authors explored important insights that may contribute to the domain of knowledge in the trends of unintended pregnancy, induced abortion, and its associated factors in Ethiopia. This may contribute to the intervention in adolescent health services in the country. However, additional work is needed to be considered for publication in its current form. For instance, the language usage may need critical proofreading since there are many language errors. In the methodology sections, there are very important clarifications. The results also need clarification. Therefore, it should address those issues before it is considered for publication.

Specific Comments

Page 1-2: Abstract

• Background: 'Although the trends of unintended pregnancy and induced abortion have declined over time in Ethiopia’ if the trend is known. What is the necessity of the trend analysis?

Authors’ response: We thank the reviewer for picking the important issue in this work. Now we have amended the part to read as determinants of unintended pregnancy and induced abortion. 

• Methods: The age category of the study participants should be clearly presented in this section. Methods in which the data collected should be described to show clarity for the readers the analysis methods should also be clear at this point.

Authors – Thanks for the valid comment. Now we have included the ages of study participants and method of data collection. [Abstract-Method part: Page 2, paragraph 1, line 2 &4]

• Results: the reduction is better if it is presented in percent (with confidence intervals), and the direction of the association should be clear with reference categories.

Authors - Thanks for the emphasis here. Yes, we have presented the decline using percent along with its CI. In addition, the negative signs are to show the factors associated with the decline over the study period. [Abstract-result part: Page 1, paragraph 2, line 3-10]

• Paragraph 3, ‘Results from the EDHS 2016 report indicate that unintended pregnancy and induced abortion are still high.' What was the comparison—high or low?

• The introduction lacks an explanation of the relevancy of the study and the value it adds to the scientific community.

Authors’ response: Thanks. This is a valid point. We have revised the introduction part and the high unintended pregnancy according to EDHS 2016 was in comparison with WHO study report as indicated in Reference number 11 [Page 3, paragraph 1, line 1-3]. Moreover, we have added some points to indicate the relevance of the study to the scientific community. [Page 4, paragraph 1, line 4-9].

Page 4-8: Methods

• ‘According to the Ethiopian census report, by the year 2018, the estimated population of Ethiopia was 106.8 million; of this, 24.2% were adolescents (from 9 to 18 years old), and 11.7% were 15 to 19 years old. Adolescent girls were 5.8% of the total population.’ References are needed for this information.

Authors – Thanks for the feedback. Now we have included reference and reads as reference #13. [Method part: Page 5, paragraph 1, line 2]

• Table 1: What were the criteria to select those variables? For me, it is important to explain the justification for selecting those variables. Since EDHS has so many variables, it may need expert consultation or other selection methods. (e.g., data mining approach)

Authors: Thanks for the comment. We used a simple Univariate filter, cross-tabulations, clinical knowledge and previous other related literature of interest; whether to decide which variables to keep in the analysis or not. [Method part: Page 6, paragraph 1, line 6-8]

• Data management: it is not clear how the incomplete data issues were addressed in this analysis.

Authors: Thanks for the valid comment. Outcome variables with missing information (information missing on unintended pregnancy and abortion) were excluded from the study whereas; due to the cross sectional nature of the DHS survey, explanatory variables with greater than 5% missing value were excluded from further analysis. [Method part: Page 7, paragraph 1, line 10-13]

Page 8-19: Results and discussion

• The result and discussion section were well written. However, language proofreading is needed.

Authors: Thanks for the valid comment

Now, we have improved the write-up and final version

Reviewer 2**************************************

Firstly, I would like to appreciate the authors for their work. my concerns are as follows:

1. there is an editorial error in some parts of your document specially duplication of words (see the abstract/introduction and method section).

Authors: - Thanks for the feedback. Now we critically revised the current version. 

2. most of the data/figures presented in the introduction section are outdated and you better rewrite it with Up-to-date data 

Authors:- We thank the reviewer for your suggestion. Now, we have presented figures from updated sources and we have highlighted these under the reference list. 

3. The discussion section is not well discussed. For example, you have mentioned that the prevalence of modern contraceptive usage in Ethiopia in 2016 as high and as this has special contribution for the reduction of induced abortion. how can you say this figure high though it is far less than half/50%?.

Authors:-Thanks for the valid comment. Now we have amended the confusing expression and modified it as relatively increased maternal health service utilization like modern contraceptive utilization (35%) in 2016 [5, 12] compared to (6%) in 2000 [12, 17, 20]. 

[Page 20, Under Discussion, Paragraph 2, line 4] 

4. the explanation given for Somali region is not convincing. You said that it could be due to NGO....is there specific NGO working in Somali region specifically working in this area?

Authors:- We thank the reviewer for raising this critical point: Now, we have improved our discussion and suggested convincing justifications. [Page 21, paragraph 1, 1-11].

5. The recommendation also lacks clarity and you better rewrite it.

 Authors: - Many thanks. Now the recommendation reads clear. [Page 22, paragraph 1 line, 6-10]

Reviewer 3**************************************

Reviewer #3: In the conclusion section, the authors recommend interventions tailored to developing regions in Ethiopia. How can we say a significant association between a factors in two emerging regions implies a problem for the four developing regions in Ethiopia? On top of this, the authors should define developing regions in Ethiopia to inform the readers.

Authors: - Apologies for the confusion we made. Now the conclusion reads clear and gives clear impression for the readers. We have also made plausible conclusion for the finding. 

[Page 22, Paragraph 1]

The conclusion is not different from the discussion section. I suggest to the authors that they revise the conclusion part.

Authors: - Apologies for that. Now the conclusion reads different 

[Page, 1 under abstract, method part, line 4-8]. 

It is better to rephrase repetitive words and phrases throughout the article. E.g., the title of the article is mentioned in the abstract and introduction. And also, in "this study," at the beginning of all paragraphs of the discussion and conclusion.

Authors: - Thank you for the vital point. Now, we made critical revision and proof reading.

---

## [Decision Letter · Decision Letter 1]

6 Feb 2024

Determinants of Unintended Pregnancy and Induced Abortion among Adolescent Women in Ethiopia: Evidence from multilevel mixed-effects decomposition analysis of 2000 - 2016 Ethiopian Demographic and Health Survey Data

PONE-D-23-09938R1

Dear Dr. Shewano,

We’re pleased to inform you that your manuscript has been judged scientifically suitable for publication and will be formally accepted for publication once it meets all outstanding technical requirements.

Within one week, you’ll receive an e-mail detailing the required amendments. When these have been addressed, you’ll receive a formal acceptance letter, and your manuscript will be scheduled for publication.

Kind regards,

Takele Gezahegn Demie, MPH

Academic Editor

PLOS ONE

Additional Editor Comments (optional):

Reviewers' comments:

Reviewer's Responses to Questions

**Comments to the Author**

1. If the authors have adequately addressed your comments raised in a previous round of review and you feel that this manuscript is now acceptable for publication, you may indicate that here to bypass the “Comments to the Author” section, enter your conflict of interest statement in the “Confidential to Editor” section, and submit your "Accept" recommendation.

Reviewer #1: All comments have been addressed

Reviewer #3: All comments have been addressed

2. Is the manuscript technically sound, and do the data support the conclusions?

Reviewer #1: Yes

Reviewer #3: Yes

3. Has the statistical analysis been performed appropriately and rigorously? 

Reviewer #1: Yes

Reviewer #3: Yes

4. Have the authors made all data underlying the findings in their manuscript fully available?

Reviewer #1: (No Response)

Reviewer #3: Yes

5. Is the manuscript presented in an intelligible fashion and written in standard English?

Reviewer #1: (No Response)

Reviewer #3: Yes

6. Review Comments to the Author

Reviewer #1: (No Response)

Reviewer #3: The Authors produce a scientifically sounded article for this important public health earea using the reliable data source.

7. PLOS authors have the option to publish the peer review history of their article (what does this mean?). If published, this will include your full peer review and any attached files.

Reviewer #1: No

Reviewer #3: **Yes: **Mamo Dereje Alemu

---

## [Editor Report · Acceptance letter]

7 Mar 2024

PONE-D-23-09938R1 

PLOS ONE

Dear Dr. Shaweno, 

I'm pleased to inform you that your manuscript has been deemed suitable for publication in PLOS ONE. Congratulations! Your manuscript is now being handed over to our production team.

Kind regards, 

on behalf of

Mr. Takele Gezahegn Demie 

Academic Editor

PLOS ONE